# Moderate spectral resolution solar irradiance measurements, aerosol optical depth, and solar transmission from 360 to 1070 nm using the refurbished Rotating Shadowband Spectroradiometer (RSS)

Joseph J. Michalsky[1,2] and Peter W. Kiedron[2]

[1]NOAA Earth System Research Laboratory, Boulder, Colorado, 80305, USA
[2]Cooperative Institute for Research in Environmental Sciences, University of Colorado, Boulder, 80309, USA

*Correspondence to*: Joseph J. Michalsky (joseph.michalsky@noaa.gov)

**Abstract.** This paper reports on a third generation rotating shadowband spectroradiometer (RSS) used to measure global and diffuse horizontal plus direct normal irradiances and transmissions at 1002 wavelengths between 360 and 1070 nm. The prism-dispersed spectral data are from the ARM Southern Great Plains site in north central Oklahoma (36.605 N, 97.486 W) and cover dates between August 2009 and February 2014. The refurbished RSS isolates the detector in a vacuum chamber with pressures near $10^{-7}$ torr. This prevents the deposition of outgassed vapors from the interior of the spectrometer shell on the cooled detector that affected the operation of the first commercial RSS. Methods for (1) ensuring the correct wavelength registration of the data and (2) deriving extraterrestrial responses over the entire spectrum, including throughout strong water vapor and oxygen bands, are described. The resulting data produced are archived as ARM data records and include cloud-screened aerosol optical depths as well as spectral irradiances and direct normal solar transmission and normalized diffuse and global irradiances.

## 1 Introduction

The rotating shadowband spectroradiometer was developed to provide spectrally-resolved measurements of the shortwave spectrum for the Atmospheric Radiation Measurement (ARM) program (Stokes and Schwartz, 1994). The instrument measures global and diffuse horizontal irradiance by alternately shading and unshading the diffuser that serves as the $2\pi$ steradian input optic to the spectrometer; it then calculates direct normal irradiance from these measurements and a laboratory-measured, spectrally-dependent correction for the angular (cosine) response of the receiver. A spectrally-dependent correction for diffuse horizontal irradiance is also made using these measured angular responses of the receiver.

To date RSS data have been used in several studies, for example; to derive water vapor by fitting a model to direct spectral irradiance data (Kiedron et al., 2001), to derive water vapor in overcast conditions by using diffuse spectral irradiance (Kiedron et al., 2003), to measure the photon pathlength to help decipher the structure of clouds over the ARM site in northern Oklahoma (Min and Harrison, 2009; Min et al., 2001; Min and Clothiaux, 2003), and to better understand aerosol retrievals (Gianelli et al., 2005) using the expanded wavelength data set.

Only the key features of the RSS will be highlighted in this paper. The first generation of visible-wavelength RSSs are briefly described in Harrison et al. (1999). Two prototypes used 512 and 1024 CCD pixel arrays and were reasonably stable. The first commercial RSS had a problem with contamination of the detector surface due to suspected outgassing from the walls of the housing surrounding the optical train, and, consequently, it was difficult to maintain calibration. This third version of the RSS fixes this problem as will be discussed in the next section. Our method to ensure correct wavelength assignments to the spectra is described and illustrated. Our approach to estimating the top-of-atmosphere responses of the instrument in the water vapor and oxygen bands, where traditional Langley analysis cannot be used, is described and demonstrated. Examples of derived quantities, including normalized irradiances, aerosol optical depths, and spectral irradiances for all three components are given in the next three sections. These results are summarized in the final section, and links to where the datasets and documentation can be found are provided.

## 2 Fundamental Instrument Details

Since the basics of the rotating shadowband spectroradiometer (RSS) that operates in the visible wavelengths and the ultraviolet version (UV-RSS) have been described in previous papers (Harrison et al., 1999; Kiedron et al., 2002, respectively), only a brief description of the key features of the RSS will be provided in this paper.

The light from the sun and/or sky passes through a diffusing disk that is designed to provide an approximate Lambertian (cosine) response to incoming radiation at all wavelengths. The light from the diffuser enters an integrating cavity that has an exit slit that passes light to the optical train discussed next. After the slit there is a shutter that is used to assess the dark counts coming from each pixel. The spectrograph contains a collimating lens followed by two prisms in tandem that achieve a moderate spectral resolution, which has a FWHM (full width at half maximum) of 0.6 nm near 360 nm and FWHM of 7 nm near 1070 nm. Pixel spacing is about 0.2 nm at the shortest wavelengths and about 2.0 nm at the longest. The chromatic aberration in this system requires that the detector, positioned after the focusing lens, be tilted to optimize the focus at all wavelengths.

The band that shadows the diffuser is positioned below the horizon at the beginning of every cycle where dark and then global horizontal samples are taken to set the integration time that depends on the irradiance level; it then moves to three positions near the sun and samples at each of these. Two of these stops are near and on either side of the sun, but do not block it; the mid-stop totally blocks direct sunlight. The sideband measurements are used to calculate a first order correction for excess skylight blocked by the band during the measurement with direct sunlight totally blocked. Using these measurements and pre-deployment, wavelength-dependent cosine response corrections, global and diffuse horizontal and direct normal irradiances can be calculated for 1002 contiguous wavelengths.

The inside of the spectrograph is maintained at a temperature near 45° C, but the detector itself is maintained at a temperature near 20° C. This provided a detector surface in the original design that was a target for condensates from outgassing vapors from the interior of the spectrograph that caused the responsivity to change over time. In the original commercial instrument, this meant frequent recalibrations to keep up with the changing response of the detector. In this new design the detector is mounted on a copper cold finger and housed in a windowed vacuum chamber that is held at a pressure of around $10^{-7}$ torr , which effectively eliminates the condensation issue (see Figure 1).

## 3 Operational Details

The refurbished RSS was deployed at the ARM site in northern Oklahoma (36.605° N, 97.486° W, 317 m) and began operating on 26 August 2009; the shutter stopped opening on 21 February 2014. Between 25 December 2013 and 21 February 2014 intermittently the shutter did not open for the measurement cycle, thus many data were lost during the last three months of operation. The cycle of four irradiance measurements plus one dark measurement was repeated every minute starting at the top of the minute. Exposure times for each measurement were between 0.4 and 4 seconds based on the irradiance level at the start of the cycle. There is a small variation in the sampling time caused by band travel to the sun's position. In the morning the sampling is nearer the top of the minute and somewhat later in the afternoon. The band motor speed is, however, about one revolution per 6 seconds, therefore, the delay in afternoon sampling is less than this.

There were some issues with band alignment, which were detected using a Fast-Fourier Transform (FFT) procedure (see, for example, Alexandrov et al. (2007)), that resulted in data being flagged as suspect. Furthermore, the initial five weeks of data showed suspicious wavelength dependencies in the AODs, perhaps caused by a very few, poorly determined extraterrestrial response retrievals at the beginning of the measurement set, and, therefore, we suggest that data taken before 1 October 2009 have a large uncertainty; however, irradiances, direct normal transmissions, and normalized diffuse and global irradiances for the first five weeks were not removed from the database and should be used with caution.

### 3.1 Wavelength Registration

Although the spectrograph is rigidly secured to the frame that stabilizes it, slight changes in pixel alignment (note that pixels are about 14 μm wide) can occur due optical shifts associated with the thermal environment or slight

mechanical movements within the instrument, for example, caused by high wind speeds. During the four and a half years of measurements, pixel shifts to shorter and longer wavelengths of up to four pixels, or about 0.8 nm at the shortest wavelengths and about 8 nm at the longest wavelengths, in either direction were noted. Since we wish to use the same wavelength coverage for the entire period, we were left with 1002 pixels whose response was deemed satisfactory over the complete period of record. This included wavelengths between 360.4 and 1070.1 nm.

To ensure proper wavelength registration, nine global horizontal spectra, typically taken near solar noon each day, were averaged. This average was compared to the global horizontal spectrum used as a standard for all days where the wavelength registration was carefully determined using pencil lamp spectra and solar absorption features. Both of these global horizontal spectra were fitted with low order spline fits that were subtracted from each to enhance the major absorption features. A cross-correlation was performed between these two residual spectra at tenth-pixel increments until the correlation between spectra reached a maximum. Only wavelengths less that 672 nm were used for the spline fit and cross correlation because these absorption features were mostly in the solar spectrum, and the terrestrial spectrum only has weak absorption lines in this part of the spectrum. We did not want to cross-correlate terrestrial features on different days where the water vapor content may have influenced the cross correlation. The spectra for that day were then shifted by the amount of pixel offset from the 'standard' spectrum. This spectral shift process is illustrated in Figure 2. Note that the blue-colored spectrum on the left is displaced slightly shortward relative to the extraterrestrial and standard spectrum (designated 'tothor' in the figure). The shift in the left panel of Figure 2 was two pixels. The process described above was then perform and the blue-colored spectrum replotted on the right. On the right all three spectra are well aligned indicating that a simple shift in the spectrum to the nearest 0.1 pixel realigns the measured spectrum to the standardized total horizontal and extraterrestrial spectra. This shift for the spectra is performed only once each day since the shifts that are observed undergo slow, subtle changes.

**3.2 Estimation of Extraterrestrial Response in Strong Terrestrial Absorption Bands**

It is our goal to generate a continuous spectrum of the direct normal transmission and normalized diffuse and global irradiances over the entire wavelength span of the RSS. These are calculated by dividing the measured response of the RSS by the response at the top of the atmosphere (TOA) adjusted for solar distance. Many portions of the solar spectrum can use the standard Langley analysis technique to estimate the TOA response (see, for example, Chapter 7 of WMO/GAW (2016)). In this method the natural logarithm of the measured response is plotted against the calculated air mass to yield estimates of the total optical depth and the response of the instrument at the top of the atmosphere. Kindel et al. (2001) used MODTRAN runs to generate model-based Langleys to demonstrate where in the spectrum Langleys were valid. Measurements in the strong bands of $H_2O$ and $O_2$ are not appropriate for standard Langley analysis, since a linear curve of growth is not expected for these strong molecular bands, and, consequently, the extrapolation to zero airmass underestimates the TOA at these wavelengths.

Reagan et al. (1987) and Bruegge et al. (1992) were among the first to perform a modified Langley analysis to derive water vapor, but as Michalsky et al. (1995) pointed out this method depends on stable water vapor over the measurement period used for the modified Langley, which is seldom observed. Consequently, a very large number of modified Langleys, which require a very long period of time and a stable instrument over that time, are typically required to even approach the accuracy in extraterrestrial response that a standard Langley analysis can achieve when working outside strong molecular bands.

Kindel et al. (2001) describe a technique for determining extraterrestrial (ET) responses over the continuous solar spectrum using spectral regions where Langley analyses are appropriate and lamp calibrations where they are not. The lamp and Langley calibrations where Langley calibrations are valid determine a scale factor applied to the lamp calibrations that is used to estimate ET responses in the spectral regions where Langleys are not appropriate. In this paper, however, we will use the method described below.

Here we used an interpolation over two strong $O_2$ bands and three strong $H_2O$ bands. The function used for the interpolation is given by equation (1)

$$V_0'(\lambda) = R(\lambda) \cdot ET(\lambda) \left[ \frac{\dfrac{V_0(\lambda_2)}{R(\lambda_2) \cdot ET(\lambda_2)} - \dfrac{V_0(\lambda_1)}{R(\lambda_1) \cdot ET(\lambda_1)}}{\lambda_2 - \lambda_2} (\lambda - \lambda_1) + \frac{V_0(\lambda_1)}{R(\lambda_1) \cdot ET(\lambda_1)} \right] \tag{1}$$

where $V_0'(\lambda)$ is the estimated extraterrestrial response in the molecular band at wavelength $\lambda$; subscripts 1 and 2 of $\lambda$ indicate $V_0(\lambda)$'s determined from standard Langley analysis, RSS responsivity $R(\lambda)$, and extraterrestrial irradiances $ET(\lambda)$ at wavelengths $\lambda_{1,2}$ just before and just after each of these five molecular bands, respectively.

Extraterrestrial (ET) solar irradiance at RSS spectral resolution was determined using the estimated slit function of the RSS applied to the high spectral resolution ET spectrum of Kurucz (http://rtweb.aer.com/solar_frame.html), but scaled to the low-resolution, but well-determined absolute ET spectrum of Gueymard (2006). Figure 3 displays the extraterrestrial spectrum at RSS spectral resolution indicating higher spectral resolution in the short wavelengths and lower resolution at long wavelengths as expected for a prism spectrograph.

The response function $R(\lambda)$ in equation (1) was determined by dividing the RSS response in counts by the calibration lamp output in $W/m^2$-nm. Figure 4 is a plot of the RSS responsivity. We only require relative response for the interpolations using equation (1).

If we divide both sides of equation (1) by $ET(\lambda)$, and if we set $R_0(\lambda) = V_0(\lambda)/ET(\lambda)$ and $R_0'(\lambda) = V_0'(\lambda)/ET(\lambda)$, then equation (1) becomes

$$R_0'(\lambda) = R(\lambda) \cdot \left\{ \left[ \frac{(R_0(\lambda_2)/R(\lambda_2) - R_0(\lambda_1)/R(\lambda_1))}{(\lambda_2 - \lambda_1)} \right] \cdot (\lambda - \lambda_1) + R_0(\lambda_1)/R(\lambda_1). \right\} \tag{2}$$

In this configuration the form of equation (2) implies that this is effectively a linear interpolation of lamp-measured response between the two wavelengths $\lambda_1, \lambda_2$ where we have valid Langley calibrations. The only variables in this
configuration are responsivity and wavelength. Therefore, given the Langley-measured response at $\lambda_1, \lambda_2$, and forcing the lamp-calibrated response at these two wavelengths to agree, we can interpolate between these two wavelengths using the scaled lamp-measured response from Figure 4. Moreover, this interpolation in response is to be preferred over an interpolation in $V_0$ because of the inherit smoothness of the response of the detector itself. Given $R_0'(\lambda)$ we then solve for $V_0'(\lambda)$ using $V_0'(\lambda) = R_0' \cdot ET(\lambda)$.


Figure 5 illustrates the $V_0$ interpolation using equation (1) when implemented for five strong molecular bands within the wavelength span of the RSS. The black line is the extraterrestrial spectrum (y-axis label); the magenta line is the scaled, uncalibrated total horizontal irradiance in counts that is our standard for wavelength registration as discussed previously; the solid green line is a scaled uncalibrated retrieval of the RSS extraterrestrial response from a morning
Langley plot with the green dots representing the estimated extraterrestrial response for the strong bands of $O_2$ and $H_2O$. The $H_\alpha$ and Na lines that are in the extraterrestrial solar spectrum are identified and appear in all three spectra. The interpolations of the extraterrestrial spectrum over the $O_2$ and $H_2O$ bands appears reasonable and will be used for extraterrestrial responses as these are considered plausible estimates. With these estimates we can now calculate the continuous normalized global and diffuse horizontal irradiances and direct normal solar spectral transmission
plus normalized global and diffuse irradiances from 360 to 1070 nm.

## 4 Direct Solar Transmission and Normalized Irradiance Calculations and Examples

As explained in the previous paragraphs, now that we have estimates for the extraterrestrial response over the entire
RSS spectral response wavelength span, we can calculate the transmission (direct normal) and normalized global and diffuse horizontal irradiances at any time. Figure 6 is a plot of direct normal transmission and normalized global and diffuse irradiances near solar noon on 27 October 2009, which was clear from horizon to horizon at this time of day. Compared to the structure at short wavelengths in the extraterrestrial spectral irradiance (see Figure 3), these quantities are extremely monotonic up to 550 nm and mostly the result of Rayleigh scattering and aerosol extinction,
although Rayleigh scattering and aerosol extinction contribute throughout the entire RSS spectral range with decreasing contributions at the longer wavelengths. Both Rayleigh scattering ($\lambda^{-4}$) and aerosol scattering (often $\sim \lambda^{-1.3}$) are wavelength dependent with the contributions falling off with increasing wavelength. This explains the lower

values at the shortest wavelengths for the **dni** and the higher values at the shortest wavelengths for the **dhi**. There is only a minor indication of imperfect RSS slit function specification that gives rise to small residuals of the very strong H and K lines of singly ionized calcium (CaII) at 393.4 and 396.9 nm in the solar spectrum. The major absorption bands of $O_2$ near 690 and 760 nm and $H_2O$ near 725, 820, and 940 nm, as noted in Figure 5, can be clearly identified. Much less obvious are the $O_2$-$O_2$ bands near 477, 577, and 630 nm, although the latter two bands are near a weak $H_2O$ water band (577 nm) or near weak $H_2O$ and $O_2$ bands (630 nm)  that complicate their identification and separation. The slight falloff in direct transmission at the longest wavelengths near the end of the RSS spectrum is mostly the result of a weak $O_2$-$O_2$ band centered near 1065 nm and, perhaps, weak water vapor absorption that will be discussed later. Somewhat less discernible is the broad Chappuis $O_3$ band centered near 610 nm. Although no retrievals will be performed in this paper, the direct normal transmission in the $H_2O$ bands can be used to estimate column $H_2O$ by running a suitable radiative transfer code at the RSS spectral resolution in transmission until an optimum match to the three $H_2O$ bands is attained. Less obvious and more uncertain is the use of the Chappuis band transmission to estimate column $O_3$. Notice that the **dni** transmission on this clear day is about 95% at wavelengths above 1000 nm with slightly higher normalized global irradiance.

Figure 7 is interesting in that this is a plot with a clear path to the sun with normalized global irradiance exceeding 100% at wavelengths greater than about 650 nm (except, of course, for the strong absorption features). **dni** is slightly lower than it was in Figure 6 and normalized diffuse horizontal irradiance is considerably higher. We can understand how values above 100% are possible if we consider Figure 8, where the vertical red line marks the time of day that the spectra in Figure 7 were measured. **dni** appears unaffected at this point in time, but a short time later clouds are clearly moving in front of the sun. Because clouds are encroaching, diffuse is enhanced by direct sunlight scattering from them, which causes enhancements in the normalized global irradiance. This is a well-recognized effect in broadband solar measurements that is discussed, for example, in Vignola et al. (2020) (see p. 21 and Figure 2.14 in that reference). Additionally, Figure 7 illustrates the wavelengths where the apparent normalized global irradiance exceeds 100%.

Figure 9(a) is a plot of the normalized irradiances for a totally overcast sky on 25 October 2009 at the ARM site in northern Oklahoma. That the sky was totally cloudy was confirmed using the total sky imager collocated at the site. Note that the **dni** is zero for all wavelengths and the normalized global horizontal irradiance is hidden by the normalized diffuse horizontal irradiance plot that should and does exactly overlay it for overcast conditions. Normalized global and normalized diffuse horizontal irradiances are equal because **ghi** is calculated using $\boldsymbol{ghi = dni \cdot \cos(\theta) + dhi}$, where $\theta$ is the angle between the sun and the zenith direction. The spectrum outside molecular absorption bands indicates a slight monotonic increase with wavelength. This increase in transmission, outside molecular bands is likely caused by the increasing surface albedo with wavelength (see, e.g., McFarlane et. al. 2011), which increases the shortwave radiation backscattered to the clouds, hence, causing them to appear brighter in the near-infrared. Although there is a slight decrease centered near 600 nm that appears to counter this suggestion that the continuum monotonically increases with wavelength, this is the broad, weak Chappuis $O_3$ band. The large $H_2O$ and $O_2$ bands identified in Figure 5 are labeled left to right starting around 690 nm. Much weaker absorption bands below 690 nm can be identified. Using Sierk et al. (2004) allows us to identify the three weak bands between 550 and 690 nm with the molecules causing the absorption features. The very weak depression around 477 nm is caused by $O_2$-$O_2$ (aka, $O_4$) absorption (Michalsky et al. 1999). The feature labeled $H_2O$ near 505 nm is a water vapor band that is identified in the HITRAN (2012) database. The Ca II H and K lines short of 400 nm appear as small residuals, perhaps, caused by a slightly imperfect specification of the RSS slit function, but this is a minor issue since these are the two strongest lines in the extraterrestrial spectrum. These residuals can also be seen in Figures 6, 7, 9, and 10. The downturn at the longest wavelengths is absorption in the $O_4$ band that is centered near at 1065 nm although the wings of a $H_2O$ bands may be influencing this part of the spectrum as well. Figure 9(b) is a similar plot on 24 October 2009 with slightly higher values for the three components. This plot shows a more pronounced Chappuis $O_3$ band centered near 600 nm, which is the broad depression in the continuum. Figure 9(c) is a spectral irradiance plot from 24 December 2009 for totally overcast skies that has a starkly different appearance than the earlier plots (Figures 9(a) and 9(b)) for overcast days. On this day at the time of this measurement the radar indicated substantial ice content in the clouds above the site. The attenuation above 1000 nm is consistent with radiance transmission spectra for ice clouds presented in LeBlanc et al. (2015); for example, compare their Figure 3(b). Consequently, there is the potential to use these spectra to recognize ice phase and, more importantly, retrieve quantitative information on ice content and size. There is clearly an incentive to study this further, although it is beyond the current focus of this paper.

**5 Aerosol Optical Depths**


Since transmissions have been calculated and are available, it should be straightforward to calculate aerosol optical depths (AODs) for the parts of the spectrum that are free of strong $H_2O$ and $O_2$ absorption bands. Since we have determined estimates for extraterrestrial responses $V_0(\lambda)$'s, and we measure responses at the surface $V(\lambda)$'s, we can calculate optical depths by solving for $\tau$ in the following


$$V(\lambda)/V_0(\lambda) = e^{-\tau(\lambda) \cdot m} \tag{2}$$

where $m$ is a known, calculated airmass, $\tau(\lambda)$ is the total optical depth, and $V_0(\lambda)$ has been adjusted for the correct solar distance for the time of observation. From the total optical depth $\tau(\lambda)$ we removed the optical depths associated with Rayleigh scattering and Chappuis band $O_3$ absorption as explained in Michalsky et al. (2001) to retrieve estimates for AODs where there are no strong molecular bands.


While Langley calibrations are ideally performed at high altitude sites, such as Mauna Loa Observatory (MLO), we have demonstrated that calibrations can be performed at less than these ideal sites using a sufficient number of Langleys. Perhaps the best example of this is from Michalsky and LeBaron (2013). Using the same multifilter rotating shadowband radiometer  (MFRSR) at MLO and in Boulder, Colorado, we obtained the same calibration in

the five aerosol channels of the MFRSR to within 0.006, worst case. Granted, the signal to noise was much better at MLO even with fewer Langley events, but the medians at both places agreed (see Figure 1 in Michalsky and LeBaron (2013)). Note that the MFRSR and the RSS use the same shadowing procedure in every respect, therefore, we expect similar results from either instrument.

Figure 10 is a log-log plot of RSS optical depth versus wavelength at 11:45 local standard time for 25 November 2009 at the central ARM site in Oklahoma with Rayleigh and ozone optical depth removed. The vertical lines are positioned at the CIMEL sunphotometer wavelengths used by AERONET in this wavelength range (https://aeronet.gsfc.nasa.gov/ (Holben et al., 2001) to measure AOD. All of these are in windows not affected by the strong $O_2$ and $H_2O$ bands. As an aside, note that if we examine this and any of our similar transmission figures

that the weak water band centered near 505 nm could have a small influence on the AOD assigned to this wavelength even though it has long been a standard wavelength for AOD measurements (WMO/GAW, 2016).

The shorter wavelengths appear noisier because the wavelength alignment is to the nearest 0.1 pixels where the spectral resolution is the highest for this instrument, the signal-to-noise ratio is lowest at these wavelengths, and the

extraterrestrial spectrum is inherently more structured at these wavelengths for a prism spectrograph. In figure 10 the black dots are the AERONET-measured aerosol optical depth for the nearest sampling to this time, which is about two and a half minutes later. From the AERONET data it appears that the AODs were stable around this time. The only wavelength where there is more than 0.01 difference is at 1020 nm.  For the 1020 nm RSS pixel an additional minor correction for water vapor, based on the work presented by Smirnov (2004, unpublished), brings the

wavelength dependence more in line with the shorter wavelength points based on the linear least-square fit to the RSS data presented as green dots. The slope (Angstrom exponent) determined by a linear fit to these RSS data is 0.52, not uncommon for late autumn aerosols. Below 400 nm the determination of extraterrestrial responses is less certain because of the higher spectral resolution and the falloff in detector response. This is likely responsible for the greater noise and non-monotonic behavior in the aerosol optical depth below these wavelengths.


Of course, many more comparisons of RSS and AERONET data could be done, but MFRSR aerosol optical depth data, which use the shadow-banding method used in the RSS, have been compared to AERONET data often with very good agreement (for example, see Figure 7, Michalsky et al., 2010).

**6 Irradiance Calculations**

Just as for the aerosol optical depth calculations, if we are given the transmission, it is straightforward to calculate the spectral irradiance for all three components. All that is required is to use the extraterrestrial (ET) spectral irradiance at the spectral resolution of the RSS (see Figure 3) and adjust for the solar distance at the time of the

measurements. Multiplying each component (**ghi, dni, and dhi**) by this distance-corrected ET spectral irradiance for each of the pixels yields the estimated spectral irradiance for each component at the surface.

Figure 11 is the calculated spectral irradiance for the same day as in Figure 3, namely, 27 October 2009 at the ARM site in northern Oklahoma. Most of the uncertainty associated with irradiance data arises from the uncertainty in the ET spectral irradiance. Transmission uncertainty in the continuum sections of the spectrum is estimated at 1% based on the analysis in Michalsky et al. (2001), but the interpolation used in the strong absorption bands results in larger uncertainties although those uncertainties are not estimated here. However, the ET uncertainty is 2% or higher, as indicated by Gueymard (2018; private communication), depending on what portion of the 360-1070 nm spectral range is under study resulting in even higher (perhaps, ~ 3% or greater) uncertainties in irradiances. Of course, for cloudy periods the **dni** will be zero and the **ghi** and **dhi** will be the same as can be inferred from Figure 9. In Figure 11 note that the **ghi** is not the sum of **dhi** and **dni** because **dni** is not the direct projected onto a horizontal surface, which would be **dni** · cos(sza).

**7 Summary**

This paper describes a vacuum system for the CCD detector that was implemented to remove issues with contamination in the first commercial version of the rotating shadowband spectroradiometer (RSS). The RSS data are used to calculate spectrally continuous global and diffuse horizontal plus direct normal irradiances; ET normalized direct, diffuse horizontal and global horizontal irradiance; and aerosol optical depth outside strong terrestrial absorption bands at many of the 1002 wavelengths between 360 and 1070 nm. The prism-dispersed spectral data are from the ARM Southern Great Plains site in north central Oklahoma (36.605 N, 97.486 W) and include dates between August 2009 and February 2014. Methods for (1) ensuring the correct spectral registration of the data and for (2) deriving extraterrestrial responses over the entire spectrum, including throughout strong water vapor and oxygen bands, are described. Examples of ET normalized direct irradiance plus normalized global and diffuse horizontal irradiances are presented, for clear, partly cloudy, and overcast conditions. A continuous aerosol optical depth example is shown and compared to AERONET data. Finally, a clear-sky example for global and diffuse horizontal and direct normal irradiance is provided.

After this paper was submitted a referee made us aware of a newly published and improved extraterrestrial irradiance (Coddington et al. 2021; with 1% uncertainties) that should improve the calculated irradiances, but the figures for this paper were not changed.

All quality-controlled data produced are archived as ARM data records and include cloud-screened aerosol optical depths as well as all three spectral irradiances plus direct normal solar transmission and normalized diffuse and global irradiances. For those interested, the data are freely available and are archived with the ARM program and can be downloaded using  https://iop.archive.arm.gov/arm-iop/0pi-data/michalsky/RSS/. Some additional notes on data quality are at ftp://aftp.cmdl.noaa.gov/user/michalsky/ in the folder 'asked_for_stuff'.

**Acknowledgments**

The authors are indebted to Jerry Berndt for designing, building, and assembling the vacuum chamber and mount for the refurbished detector and his tenacity in making the system function. Scott Stierle not only developed software to ingest the data and provide a clean documented file for each day of the 4.5 years of RSS data, but helped deploy and performed field repairs during the  deployment. Patrick Disterhoft provided laboratory facilities and technical direction in the refurbishment of the RSS. This research was funded by the United States Department of Energy, Office of Science's Atmospheric Radiation Measurement (ARM) program. We thank the staff at NASA AERONET and the ARM program staff, especially Rick Wagener, for establishing and maintaining the AERONET site data used in this investigation. The authors are indebted to the referees of this paper who made many insightful comments that strengthened the manuscript.

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

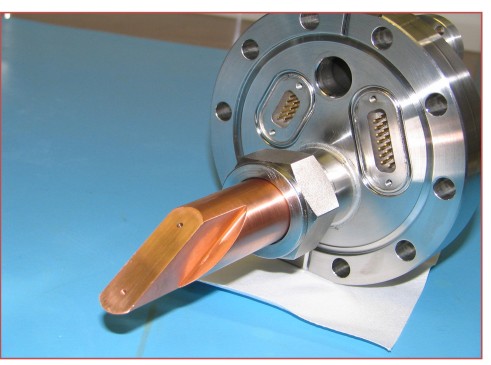
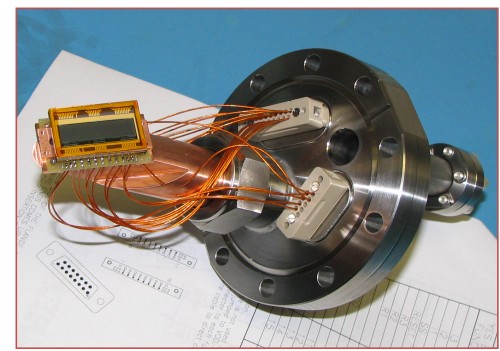
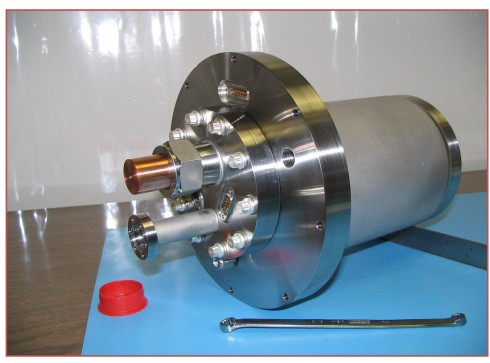
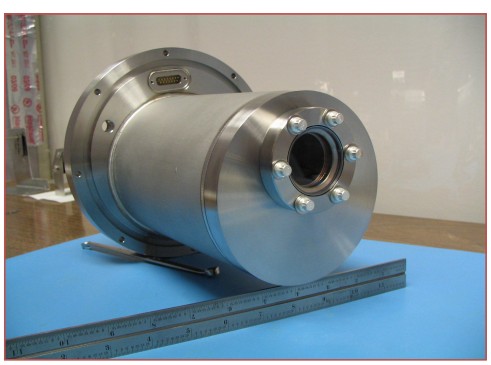

**Figure 1: Top left is the cold finger that is held at 20º C. Top right is the CCD detector that mounts on the copper cold finger. Bottom left and right are the back and front of the assembled housing. The CCD detector is behind the window.**


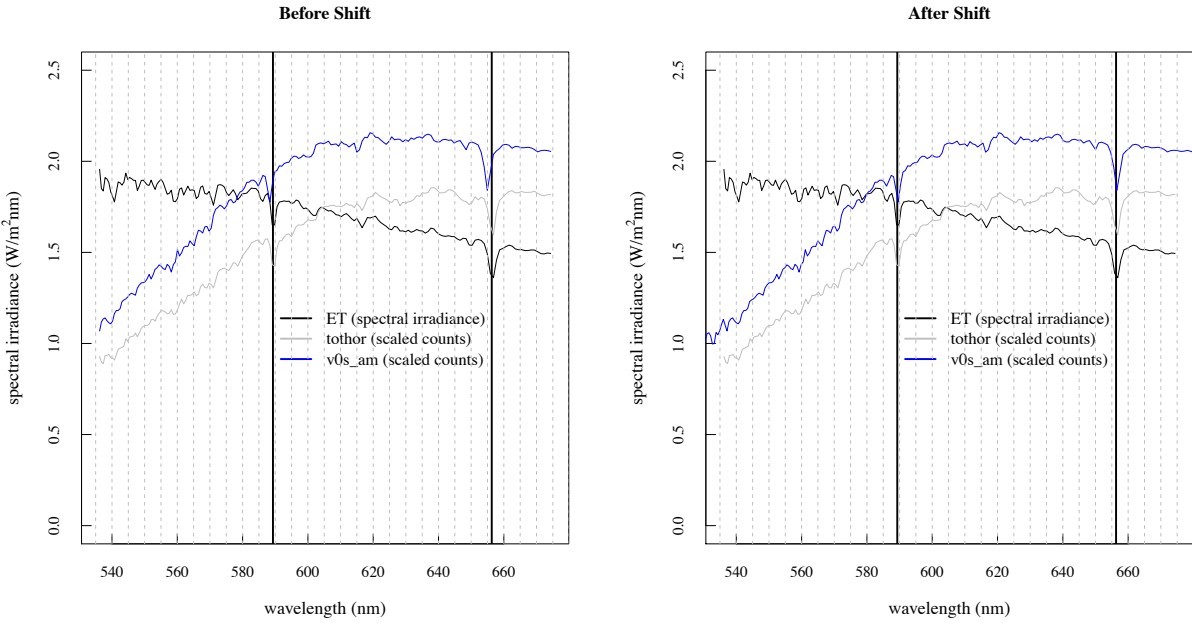

**Figure 2: The average of nine spectra near solar noon appear as the blue line; the extraterrestrial spectrum (ET) at RSS resolution is the black line (units on y-axis refer to the ET), and the gray line is the carefully determined spectrum used as the standard total horizontal spectrum that is compare to all RSS data. The blue spectrum in the left panel is shifted shortward by two pixels and the right panel shows the shift-corrected spectrum. The two absorption lines are in the solar spectrum since they also appear in the ET spectrum; these are the sodium doublet 589.0/589.6 nm and the hydrogen alpha line 656.3 nm.**

**ET Spectrum at RSS Resolution**

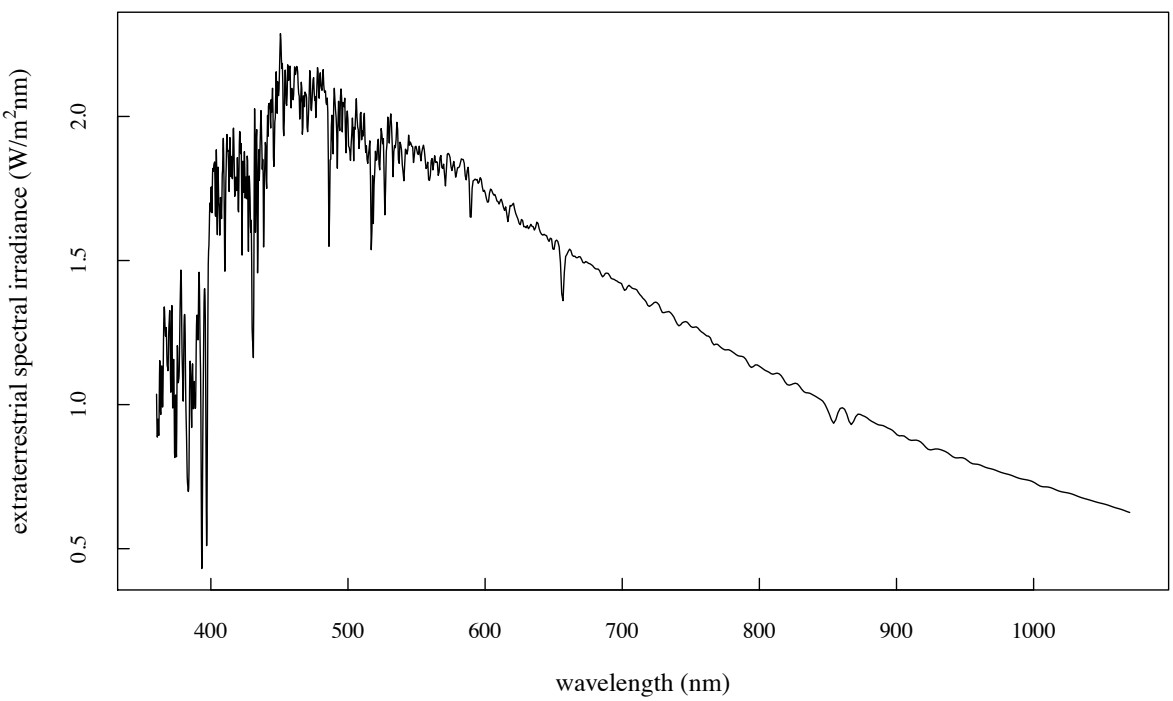

**Figure 3: Extraterrestrial solar spectrum used for this paper using RSS slit function applied to Kurucz high-resolution calculated ET solar spectrum, then scaled to the Gueymard low resolution ET solar spectrum, which is regarded as our best estimate of absolute irradiances.**


**RSS Response Function**

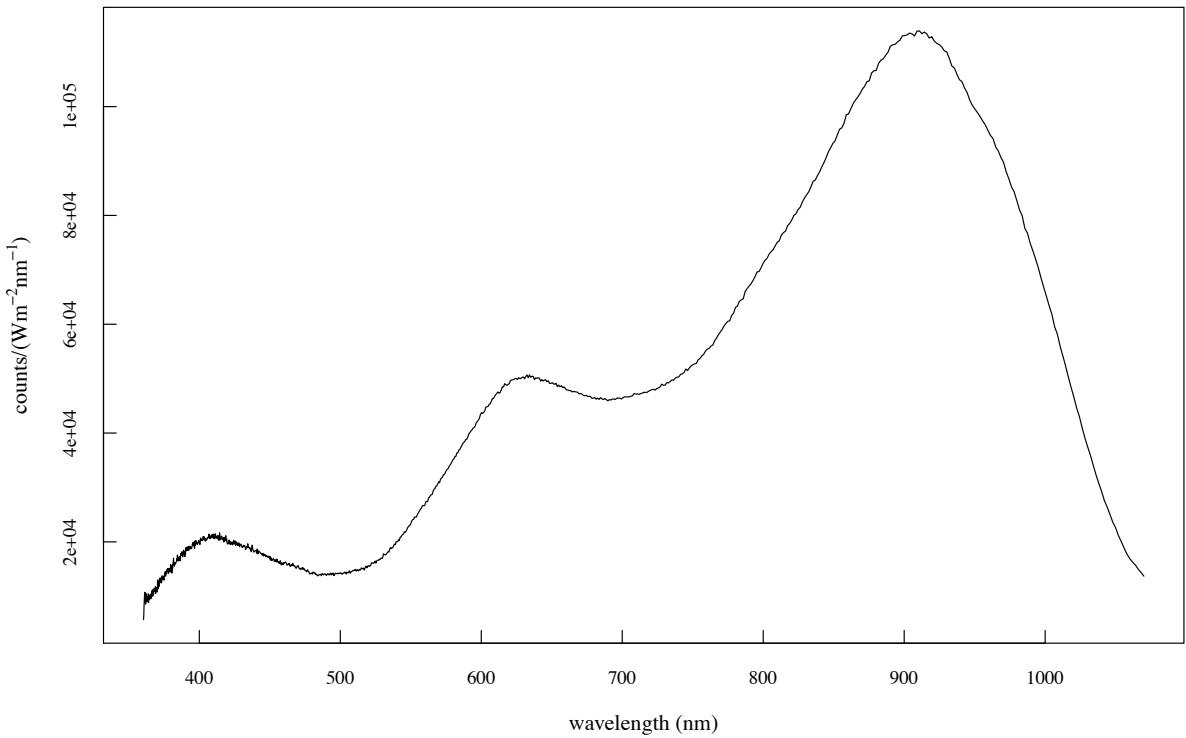

**Figure 4: Measured relative response of the RSS that is needed for the interpolation scheme represented by equation (1).**

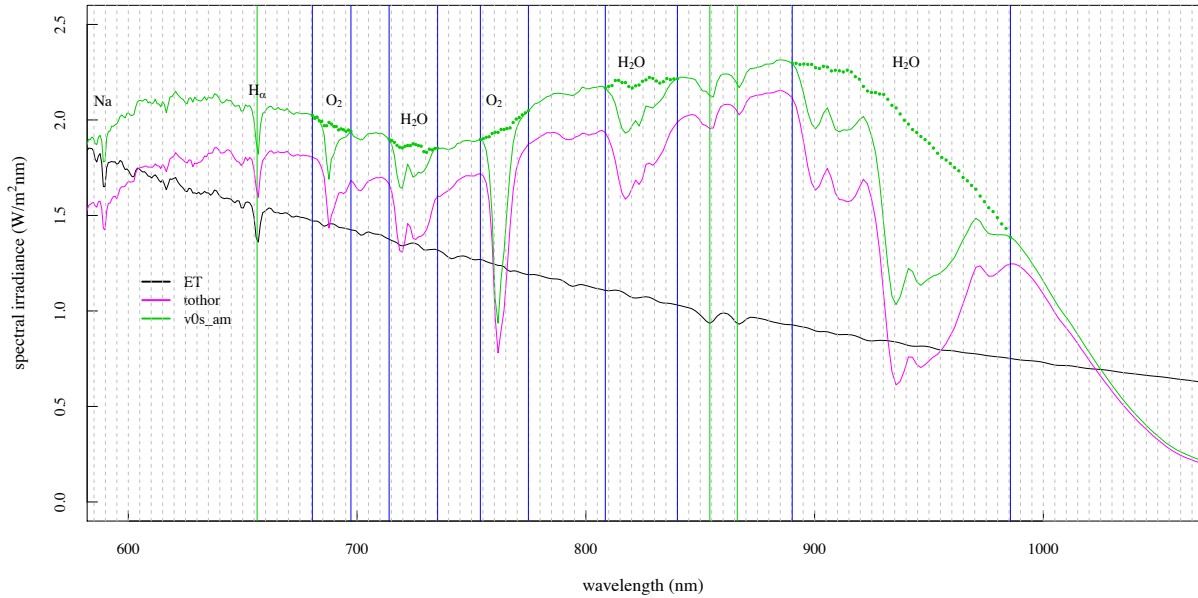

**Figure 5: Illustration of the v0 interpolation over strong $O_2$ and $H_2O$ molecular bands using equation (1) (green dots are interpolated points). Black "ET" line is the extraterrestrial irradiance (y-axis label) for mean distance to the Sun. The tothor and v0s_am are scaled to fit the plot.**

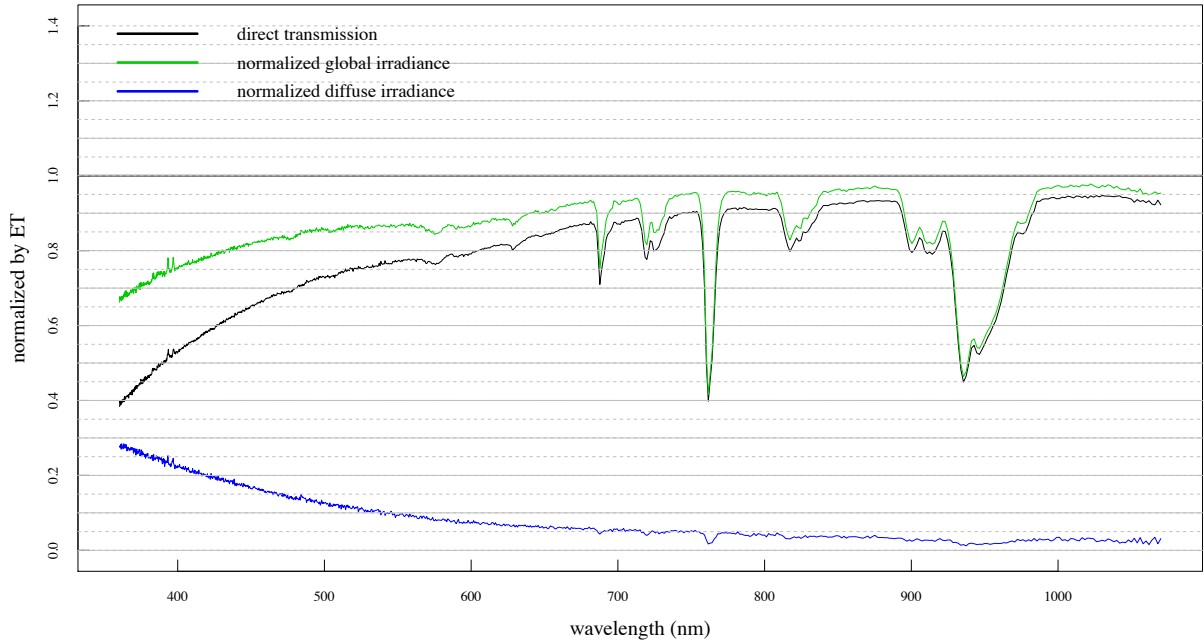

**Figure 6: Direct transmission and normalized global and diffuse irradiances on a clear day 27 October 2009 at the ARM site in northern Oklahoma. Note that both direct normal transmission and the horizontal component of direct normal transmission, which is dni * cos(solar-zenith angle), would have the same transmission profile. Note, also, that the direct transmission plus the normalized diffuse irradiance sum to the normalized global irradiance.**


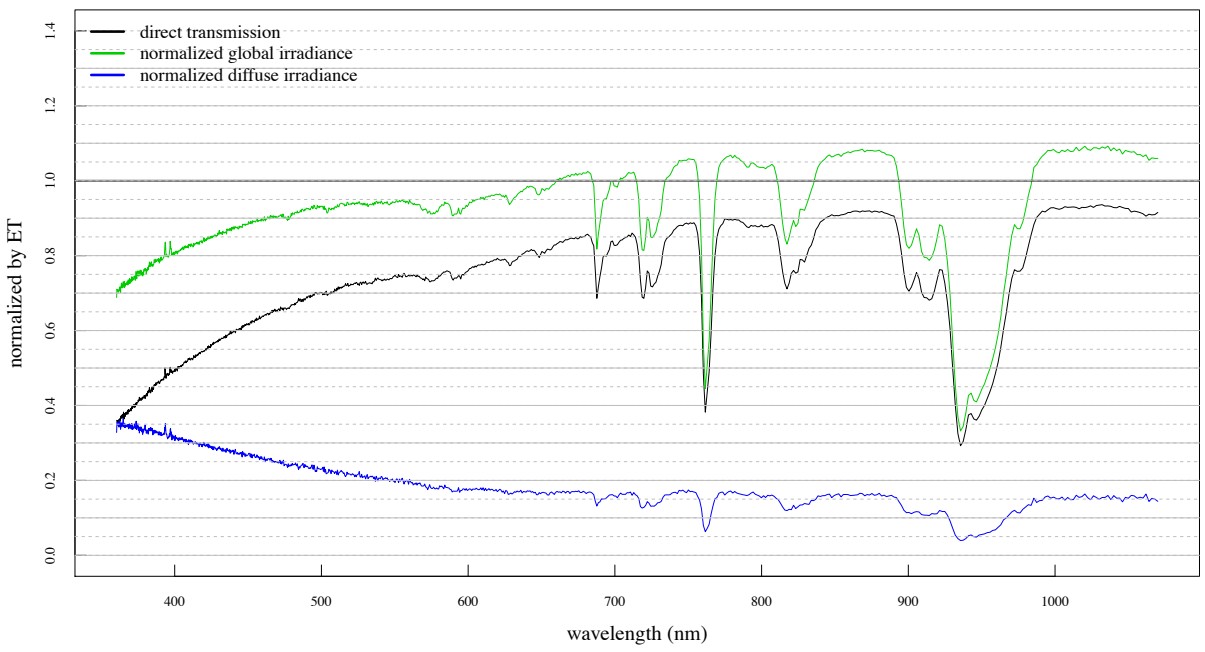

**Figure 7: RSS spectra for an instance where there is a clear path to the sun on a partly cloudy day. Note that normalized global irradiance exceeds 100% at many of the longer wavelengths.**

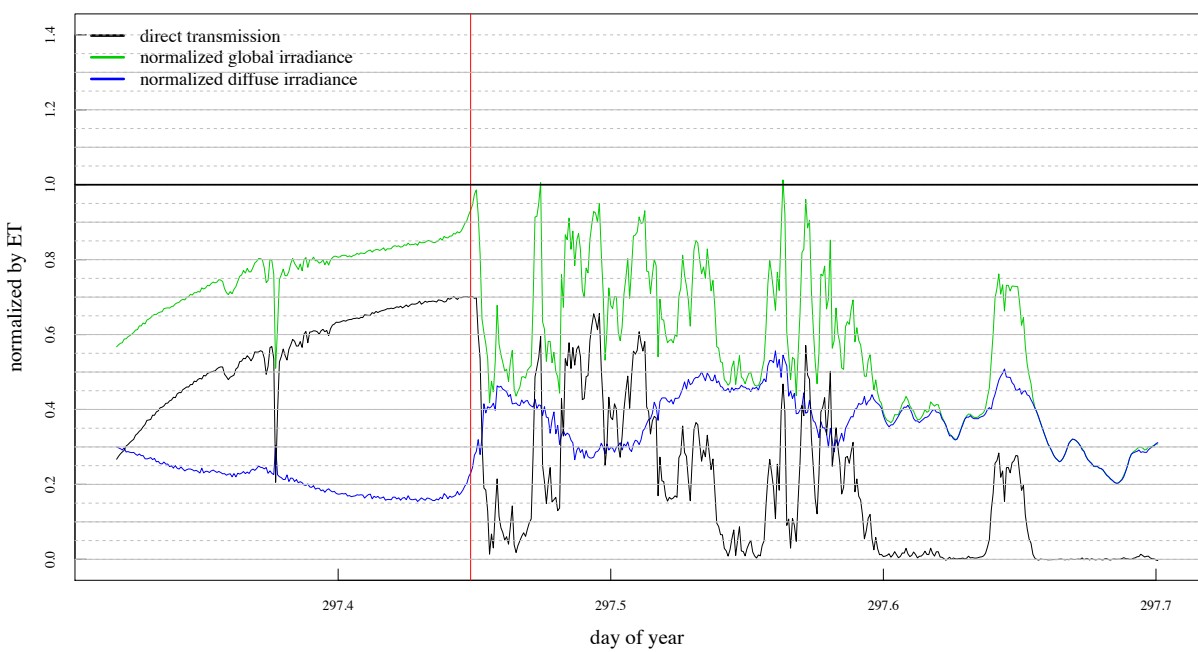

**Figure 8: Plot of the direct normal transmission and the normalized global and diffuse irradiances near 500 nm for 24 October 2009 at the ARM site in northern Oklahoma. The vertical red line marks the time of the measured spectra in Figure 7. The direct transmission appears unaffected by nearby clouds, but the normalized diffuse irradiance and, therefore, the normalized global diffuse irradiance are enhanced over the clear-sky values explaining the effect of apparent normalized global irradiance greater than 100% in Figure 7.**


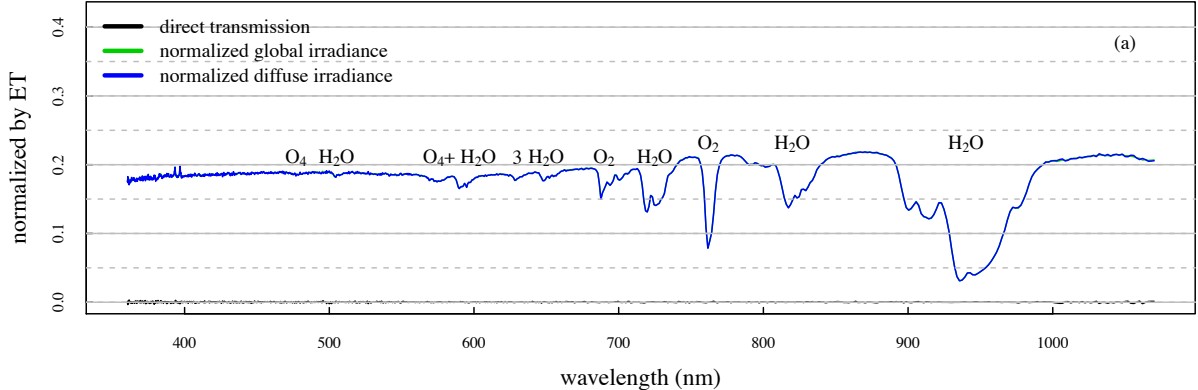

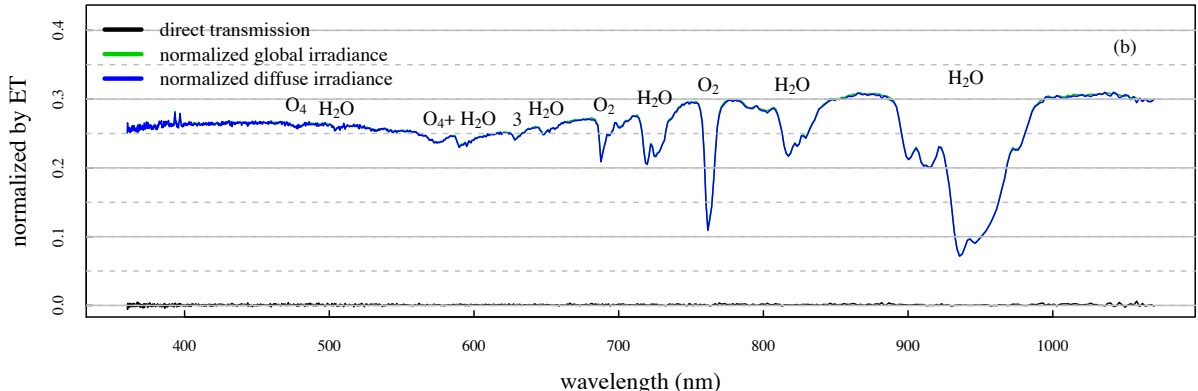

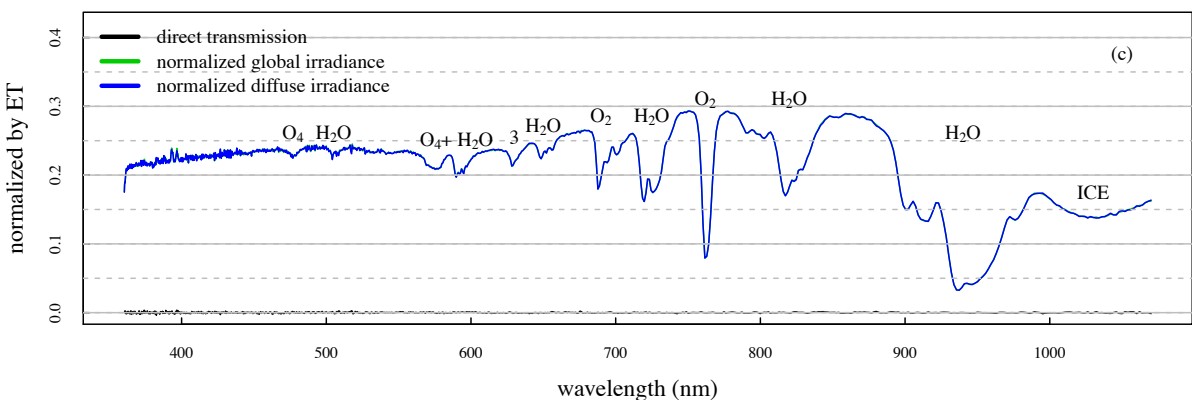

**Figure 9: (a) Spectrally continuous transmission spectrum for a totally overcast sky on 25 October 2009 at the northern Oklahoma ARM site. Outside the molecular bands there is an overall monotonic increase in transmission with wavelength. The band identified by the number "3" near 630 nm includes overlapping H₂O, O₂, and O₄ bands. The bands labeled O₄ + H₂O are overlapping bands. H₂O near 505 nm is identified as a weak water band in the HITRAN database; (b) Similar plot to Figure 9(a), but for overcast skies on 24 October 2009 with higher overall normalized global/diffuse irradiance. Note the same details for the absorption features except for the more pronounced Chappuis ozone band centered near 600 nm; (c) Spectral irradiance plot for overcast day 24 December 2009 with ice clouds present as identified**


**from ARM radar measurements. Note the pronounced difference in the spectrum beyond 1000 nm compared to Figures 9(a) and 9(b) indicated by the 'ICE' label between 1000 and 1100 nm.**

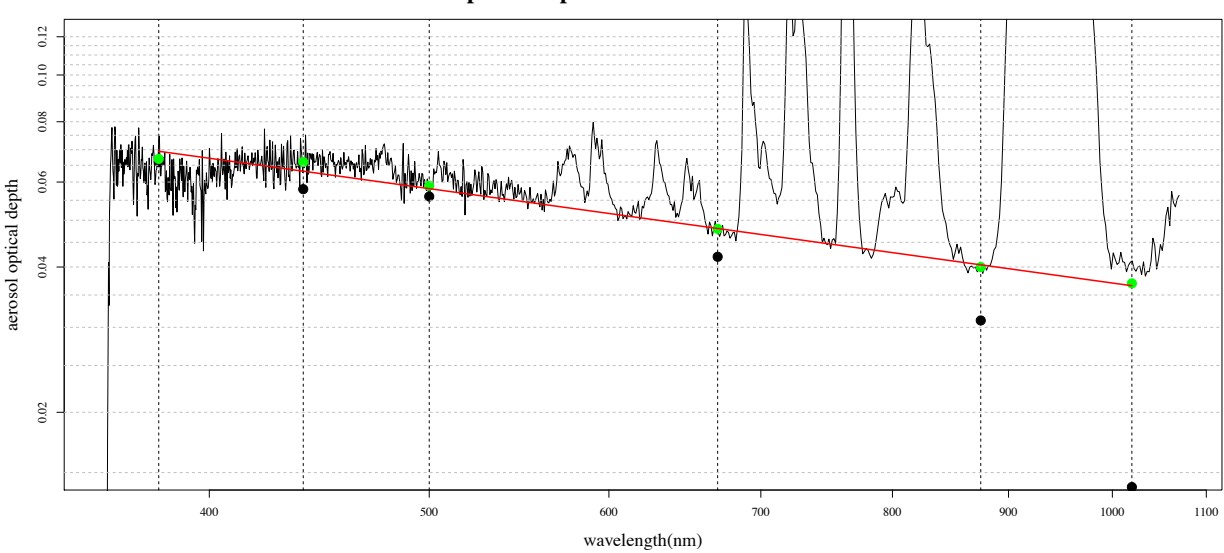

**Figure 10: RSS aerosol optical depth versus wavelength at 11:45 local standard time at the ARM central facility in northern Oklahoma. The vertical lines are drawn at the wavelengths used in the CIMEL sunphotometers employed by AERONET for measuring AODs in this part of the spectrum. The $O_2$ and $H_2O$ bands have been truncated at 0.12 optical depth to better see the aerosol wavelength dependence. Approximated RSS AOD (green dots) versus wavelength for pixels at AERONET wavelengths. Least squares fit to the RSS points yields an Angstrom exponent equal to 0.52. The filled black dots are the AERONET points taken 2.5 minutes after the RSS data. Except for 1020 agreement is within 0.01 AOD. The 1020 nm (green) point for the RSS is lower by 0.003 because an additional correction was made for water vapor at that wavelength.**

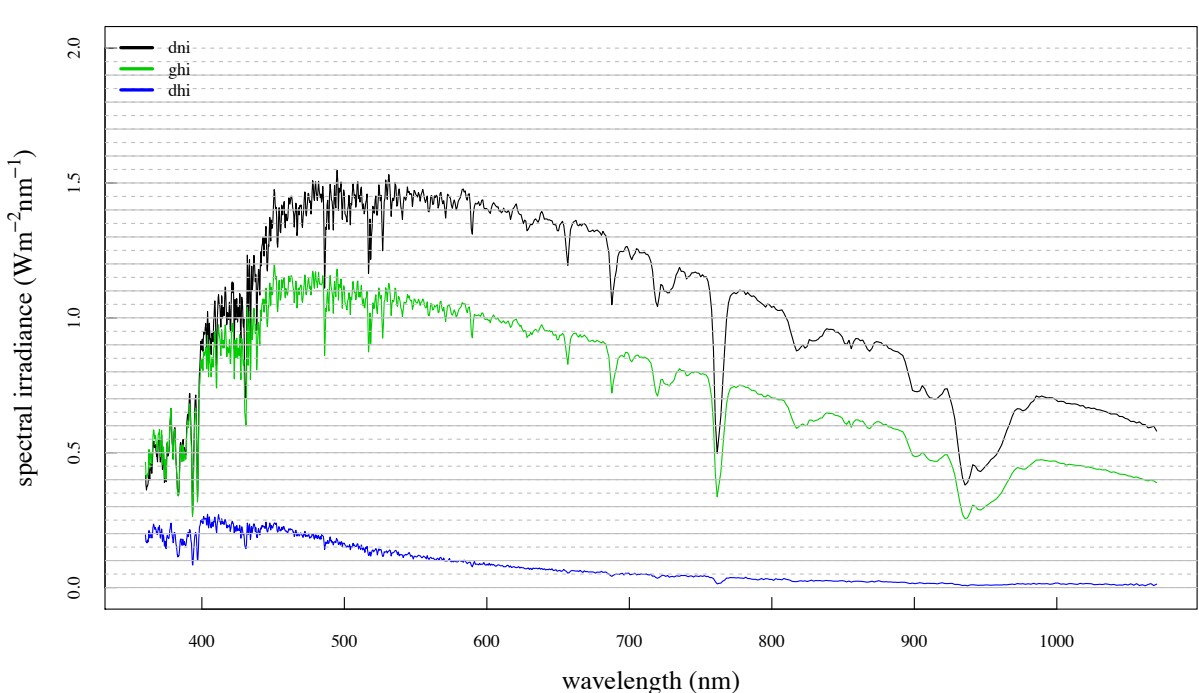

**Day of Year = 300**

**Figure 11: Spectral irradiance calculated using RSS measurements for three solar components near solar noon at the ARM site in northern Oklahoma on 27 October 2009.**