# Peer review of "Moderate spectral resolution solar irradiance measurements, aerosol optical depth, and solar transmission from 360 to 1070 nm using the refurbished Rotating Shadowband Spectroradiometer (RSS)"

_Atmospheric Measurement Techniques, 2021_

## Referee Comment (RC2)

This paper reports on the latest version of the Rotating Shadowband Spectroradiometer (RSS) which operates at the ARM site in Oklahoma making spectrally resolved measurements of downwelling solar irradiance in the 360 to 1070 nm wavelength range. The instrument has been upgraded to eliminate deposition on the detector array that degraded the radiometric performance. Additionally, the work describes a technique to radiometrically calibrate spectral regions of strong absorption (e.g. oxygen and water vapor) where Langley analysis to determine the top-of-atmosphere irradiance (the so-called V0) is not valid. The paper also reports on the correction of the spectral calibration (registration) with the use of solar atmospheric absorption features. Finally example spectra of: direct normal, global and diffuse horizontal, transmission, aerosol optical depth spectra are presented. I have several comments and suggestions that should be addressed prior to the publication of this work.

Major Comments:

1) The section 3.2 closely follows the technique demonstrated by Kindel et al (Applied Optics, 2001) using a hybrid of Langley channels in the spectral regions where transmission follows Beer's law and a standard lamp to fill in the regions of strong molecular absorption. In that work, a wavelength independent scaling factor was used to account for geometry and bring the calibration coefficients from Langley and standard lamp into agreement. This had the dual purpose of filling in spectral regions where Langley is invalid (as is done in this work) and demonstrating, to level of uncertainty of TOA solar spectra, about 2-3% and at the time of publication, the agreement between the two methods (standard lamp and Langley) for radiometric calibration of solar spectroradiometers. I am surprised, given the similarity of the technique described here that no mention or description of that work is given. It should be.

2) Langley analysis is usually performed at high altitude sites where relatively pristine atmospheric conditions prevail. The Mauna Loa Observatory is often used in the community for just such calibration experiments. No doubt, this instrument is fixed at the ARM site and the authors are left with using the ARM site for the Langley calibration. This will introduce higher uncertainties. It is impossible to determine what those uncertainties might be from this paper. Ideally, the V0s should be stable over the course of many Langley plots. A stable instrument, combined with good Langley conditions will reproduce the same V0s from Langley plot to Langley plot. To give the reader an understanding of just how good the instrument and the ARM site are for Langley analysis I recommend plotting the V0s with their standard deviations. I worry that the aerosol environment of the ARM site is less than ideal. As the authors point out, creating a valid (modified) Langley plot for water vapor requires that water vapor is stable over the course of the Langley plot measurement time frame. This is equally true of aerosols, although likely not as problematic as water vapor. Additionally, the V0s generally have higher uncertainties in the shorter wavelengths. The higher optical depths likely encountered at the ARM site will generate larger slopes that create larger changes in the V0s when extrapolated to the TOA.

3) The authors carefully identify the source of nearly every absorption feature in the spectra shown. However, there is an effect that is not discussed at all. The so-called planetary problem. This is the result of multiple scattering between the surface and the atmosphere. In measurements of downwelling global spectral irradiance taken from aircraft it's common to see the signature of the vegetation near infrared edge (~690-800 nm) when flying below a cloud deck. Indeed, the first surface measurements made with an ASD-FR spectroradiometer and cosine diffuser showed this effect and it remained a mystery until later explained by Dr. Peter Pilewskie who made similar measurements with the Solar Spectral Flux Radiometer. Thankfully, over the ocean this effect is largely absent  thus the strong preference for using flight data over ocean for shortwave cloud studies. I strongly suspect that this type of effect on the spectral shape of the irradiance can be found in RSS data. I understand that it's extremely hard to quantify without good measurements of surface spectral albedo and even then I'm not sure how those measurements would be used to correct the data. The authors should, at the very least, acknowledge the unknown continuum effect that the surface albedo has on the RSS measurements, particularly in the presence of clouds, perhaps showing a plot of a global irradiance spectrum obtained during peak growing season under cloud cover. The feature is generally easy to identify.
4) The comparison of aerosol optical depths from RSS and the well-established Cimel instrument (Figure 13) are very useful and give confidence to the results obtained by RSS. However, a single comparison, as good as it is, it not useful. Surely, this figure can be expanded so that statistics can be calculated to see how well the two instruments agree.  Perhaps aerosol optical depths from both instruments on those days used for the Langley analysis could be included?
5) There is no summary/conclusions section to this work. I often find this, after the abstract, to be the most important section in any scientific paper.

Minor Comments:
1) The use of Kurucz/Gueymard is outdated. The TSIS Hybrid Solar Reference Spectrum has uncertainties substantially below 1% for most of the wavelength range discussed in this work. (Coddington, 2021). I understand that this comes too late for this work, but I would encourage the authors to reference this work as a potential fix for their comment on line 265 about the TOA solar spectrum uncertainty.   Additionally, the Kurucz spectrum is scaled by Gueymard. It's not clear how. Is it normalized at a single wavelength or by their integrated values? https://doi.org/10.1029/2020GL091709
2) Line 207-208, I'm not sure what is meant by the sentence "The feature labeled $H_2O$(?) is likely a water vapor band that is in the HITRAN (2012) database". The feature is in the spectrum and I assume all the features described in the paper are in the HITRAN database or atlases of solar spectra. Please reword.
3) Line 213-214. "… is the broad depression in what looks like continuum". Chappuis absorption doesn't look like continuum, it is continuum. Reword.
4) Line 122 (Kindel,  2001) used synthetic Langley plots created with MODTRAN to show exactly where in the spectrum(400-2500 nm) Langley (i.e. Beer's Law) is valid.

5) Line 233, please explain how Rayleigh and Chappuis were removed, it has a large effect on the values of the aerosol optical depth. On line 183, the retrieval of ozone from the Chappuis is "less obvious and more uncertain". Certainly this applies to its removal to determine aerosol optical depth as well?

6) The transmission uncertainty is estimated at 1% on line 265. Following the comments above about the lack of uncertainty analysis in the generation of the V0s it's impossible to assess this claim.

---

## Author Comment (AC1)

Comment on amt-2021-162
Anonymous Referee #1
Referee comment on "Moderate spectral resolution solar irradiance measurements, aerosol optical depth, and solar transmission from 360 to 1070 nm using the refurbished Rotating Shadowband Spectroradiometer (RSS)" by Joseph J. Michalsky and Peter W. Kiedron, Atmos. Meas. Tech. Discuss., https://doi.org/10.5194/amt-2021-162-RC1, 2021

General comments:
The submission "Moderate spectral resolution solar irradiance measurements, aerosol optical depth, and solar transmission from 360 to 1070 nm using the refurbished Rotating Shadowband Spectroradiometer (RSS)" by Michasky and Kiedron briefly describes the third version of the RSS instrument which operated for several years at the ARM SGP central facility. The submission references previously published work for detailed descriptions of the RSS optical configuration and shadowband operation, with the scope of the current work focusing on specific modifications to the 3rd RSS implemented to mitigate issues with the previous version, followed by a discussion of details related to operational processing of the instrument data to yield quantities of interest to atmospheric scientists including hypeerspectral aerosol optical depth and irradiances.
The operational processing largely follows established methods with the exception of the approach used to determine wavelength registration and the interpolation technique introduced to infer calibrations over wavelength regions for which Langley calibrations are not valid, e.g. water vapor, oxygen bands,etc.

Specific comments -
1 Introduction:
Consider adding 1-2 sentences in the third paragraph to very briefly introduce/identify RSS #1, #2, and #3 which would allow you to eliminate the phrase "the first commercial, i.e., second generation, RSS" which I found to be rather awkward.

**[Wording changed to: "The first generation of visible-wavelength RSS's are briefly described in Harrison et al. (1999). Two prototypes used 512 and 1024 CCD pixel arrays and were reasonably stable. The first commercial RSS had a problem with contamination of the detector surface due to suspected outgassing from the walls of the housing surrounding the optical train. The third version of the RSS fixes this problem as will be discussed in the next section."]**

2 Fundamental Instrument Details:
The organization of the first paragraph could be improved by introducing each element in the order they would appear from the perspective of the light path. So, start with fore- optic (diffuser and band), integrating cavity, exit slit, prisms, detectors. (Where is the shutter? Without re-reading previous papers I can't remember where it is located in this sequence. It should be described in this section, along with the acquisition of darks, TH, SB1, BK, SB2.)
line 51 notes: FWHM (full width at half maximum) of 0.6 nm near 360 nm and FWHM of 7 nm near 1070 nm. OK, but how does the pixel spacing vary with wavelength? The pixel spacing and the spectral resolution are distint properties and both may vary with pixel. Suppose pixel A has center at 360 nm. What is the center wavelength of pixel A+1? Similarly, if Pixel Z has center at 1070 nm, what is the center wavelength of pixel Z-1?

**[Changes to the text have been made: "After the slit there is a shutter that is used to assess the dark counts coming from each pixel. The spectrograph contains a collimating lens followed by two prisms in tandem that achieve a moderate spectral resolution, which has a FWHM (full width at half maximum) of 0.6 nm near 360 nm and FWHM of 7 nm near 1070 nm. Pixel spacing is about 0.2 nm at the shortest wavelengths and about 2.0 nm at the longest. The chromatic aberration in this system requires that the detector, positioned after the focusing lens, be tilted to optimize the focus at all wavelengths."]**

3 Operational Details:
Seems as though dark subtraction shouldn't be necessary for direct beam, right? So even while the shutter was intermittently operating you should still have valid direct beam measurements, unless the shutter position was varying throughout the banding measurements.
Probably should include a reference to Mikhail Alexandrov's paper on the FFT technique to identify band issues.

**[Response: Shutter did not open, so comment added to this effect; also Alexandrov reference added.]**

3.1 Wavelength Registration
lines 96-97 >> pixel shifts ... of up to four pixels in either direction were noted
Earlier, you note the the spectral resolution varies from 0.6 nm to 7 nm from one end of the spectrum to the other, but now you're talking about the wavelength shift in terms of "pixels" and without knowing the pixel spacing it is unclear whether four pixels is 2.4 nm, 28 nm, or some other value. Can you estimate the effect in nm?

**[Done: About 0.8 nm at shortest wavelength and 8 nm at longest; this comment added to text]**

I definitely like the approach of using the average of noon-time global horizontal spectra (lowest airmass) for wavelength registration, but you haven't convinced me that a wavelength shift (as opposed to a stretch and shift) is adequate. Frequently, wavelength registration incorporates both a stretch (scale factor) and shift (offset). This might be even more important in the case of the RSS where the spectral resolution changes by more than an order of magnitude from short to long wavelength.

**[Response: We found that only a shift was necessary.]**

3.2 Estimation of Extraterrestrial Response in Strong Terrestrial Absorption Bands
I have significant concerns about this section.
1. Maybe you should include a reference for standard Langley calibration, and perhaps a 1-2 sentence description? The references provided in this section are for the "modified" Langley, but equation (1) subsumes generation of initial Vo values (for "good" Langley regions).

**[This added: "Many portions of the solar spectrum can use the standard Langley analysis technique to estimate the TOA response (see, for example, Chapter 7 of WMO/GAW (2016)). In this method the natural logarithm of the measured response is plotted against the calculated air mass to yield estimates of the total optical depth and the response of the instrument at the top of the atmosphere. Kindel et al. (2001) used MODTRAN runs to generate artificial Langleys to demonstrate where in the spectrum Langleys were valid."]**

2. Equation (1) lists Vo' on both sides of the equation. I'm pretty sure that all of the instances on the right-hand side of the equation Vo should replace Vo'.

**[Fixed]**

3. Line 139 "RSS responses R(lambda)" Do you mean "responsivity"?

**[Fixed]**

4. I don't think the responsivity R(L) is correctly described in lines 148-149. Dividing the lamp output in W/m2-nm by the calibration in W/m^2-nm/count would yield a responsivity with units of counts. According to wikipedia (https://en.wikipedia.org/wiki/Responsivity) the responsivity, in the specific case of a photodetector, measures the electrical output per optical input. So, in the case of the RSS the responsivity should be in units of counts/[W/m2-nm].

**[Fixed]**

5. I think equation (1) confuses rather than clarifies the interpolation process. Substituting "m" for the complicated looking fraction in front of $(\lambda - \lambda 1)$ and "b" for the term at the very end of the equation makes it clear that this is nothing other than a linear interpolation from $\lambda 1$ to $\lambda 2$. But what is being interpolated? Despite appearances, it is not really interpolating terms of Vo. It is really an interpolation in terms of responsivity R. You can see this by simply dividing both sides of Eq(1) by ET and defining Ro = Vo/ET and Ro' = Vo'/ET. Equation (1) then becomes:
$Ro(\lambda)' = R(\lambda) * \{[(Ro(\lambda 2)/R(\lambda 2) - Ro(\lambda 1)/R(\lambda 1))/(\lambda 2-\lambda 1)]*(\lambda-\lambda 1) + Ro(\lambda 1)/R(\lambda 1)\}$
This is much cleaner. The only terms that are left are responsivity and lambda. Physically interpolation in responsivity this makes sense because the responsivity is a material property of the detector itself, so whether determined from calibrated lamp output or from solar irradiance the quantities Ro(L) and R(L) should agree whenever the Langley calibration is valid. Thus for wavelength regions between $\lambda 1$ and $\lambda 2$ (within gas absorption bands), one interpolates the shape of the lamp-measured $R(\lambda)$ between the values of the Langley measured $Ro(\lambda)$

across that wavelength range. Mathematically interpolation in responsivity rather than Vo is also preferable becuase responsivity will naturally be smoother since solar and atmospheric features are eliminated or reduced. Then, one obtains Vo'(λ) from Vo'(λ) = Ro'(λ)/ET(λ).

**[Added text: If we divide both sides of equation (1) by $ET(\lambda)$, and if we set $R_0(\lambda) = V_0(\lambda)/ET(\lambda)$ and $R'_0(\lambda) = V'_0(\lambda)/ET(\lambda)$, then equation (1) becomes**

$$R'_0(\lambda) = R(\lambda) \cdot \left\{ \left[ \frac{(R_0(\lambda_2)/R(\lambda_2) - R_0(\lambda_1)/R(\lambda_1))}{(\lambda_2 - \lambda_1)} \right] \cdot (\lambda - \lambda_1) + R_0(\lambda_1)/R(\lambda_1). \right\} \tag{2}$$

**In this configuration the form of equation (2) implies that this is effectively a linear interpolation of lamp-measured response between the two wavelengths $\lambda_1, \lambda_2$ where we have valid Langley calibrations. The only variables in this configuration are responsivity and wavelength. Therefore, given the Langley-measured response at $\lambda_1, \lambda_2$, and forcing the lamp-calibrated response at these two wavelengths to agree, we can interpolate between these two wavelengths using the scaled lamp-measured response from Figure 4. Moreover, this interpolation in response is to be preferred over an interpolation in $V_0$ because of the inherit smoothness of the response of the detector itself. Given $R'_0(\lambda)$ we then solve for $V'_0(\lambda)$ using $V'_0(\lambda) = R'_0 \cdot ET(\lambda).$]**

4 Solar Transmission Calculations and Examples
Is an effective cosine-correction being applied to the diffuse hemispheric? If so, it might be good to say so somewhere. Relatedly, in truth aren't only two components truly independent? That is, you're computing ghi = dni + dhi, right? Probably this should be mentioned explicitly. And in later figures in overcast conditions, when dni = 0 it is neither surprising nor a measure of validity that dhi and ghi agree so well. It is a necessary consequence of the fact that you're computing ghi as the sum of dni and dhi, correct?

**[Cosine responses applied to direct and diffuse and so stated in the text.]**

Moving on, I do have a more fundamental issue. I disagree that the dhi and ghi terms are properly termed "transmission" or "transmittances". It is true that dividing the direct normal measured component V(t,λ) by Vo(λ) (or Vo'(λ)) yields the slant-path atmospheric tranmittance T(t,λ), but transmittance when computed in this way is implicitly a one-stream property. However, the measured diffuse irradiance is not a one-stream quantity. It must be either multiply scattered or due to a different incident ray (a different stream) than that used to define the direct beam transmittance. Aeronet normalizes their measurements of narrow FOV radiances by dividing by ET. They refer to these quantities as "normalized radiance", not "transmittance". I think a similar distinction is important in this case. For example, you might consider using the terms normalized diffuse irradiance and normalized global irradiance instead of referring to these as "transmittances". Not only this this terminology more accurate, it also helps explain the seeming conundrum of ghi > 1. When the sky is horizontally homogeneous, the radiation stream for different parts of the sky are essentialy the same, so one naturally expects dni and dhi to behave like a conserved sum with dni + dhi = ghi < 1. However, under broken (inhomogeneous) skies, it is possible for the direct line of sight to the sun to be cloud-free, while bright clouds away from the line of sight scatter sunlight into the diffuse hemispheric component such that dni + dhi = ghi > 1.

**[Text and figure labels changed to use 'normalized global and diffuse horizontal irradiance']**

I agree with the authors that instances with ghi > 1 are physically possible and I agree with their explanation of how it occurs. I just disagree with the terminology. As a side note, the ratio between the direct and diffuse components is a calibration indepent quantity that has found use aerosol retrievals, but in general these retrievals require horizontally homogeneous conditions. It would seem that the normalized global hemispheric component (which would also be calibration independent) might be useful to identify spatially inhomogeneous conditions.

The authors note that in Figure 9 teh Ca II, H, and K lines appear as small residuals due to imperfect wavelength registration. These features are also apparent in Figures 6 and 7, btw. However, while the scale of images makes it difficult to assess, the authors explanation seems incorrect. Imperfect wavelength registration would yield a "saw-tooth" in the vicinity of the sharp peak with an enhancement on one side and a reduction on the opposite side.

Rather, this small residual may point to an issue with the underlying spectrocopy in the ET spectrum, or more likely slight to inaccuracy in the lineshape of the RSS spectrometer.

**[We indicate that it may be that the slit function specification for the RSS may be slightly off.]**

Figures 9,10,11: Replace y axis label of "transmission" with "normalized by ET". Fix caption to avoid referring to global and diffuse components as "transmission" because they're not. Also, since dni=0, and dni + dhi = ghi, then dhi is equal to ghi, so you can plot either dhi or ghi and elimnate the other two. Then, condense figures 9, 10, & 11 into one figure to better show the similarity on days with water clouds and the contrast to the day with cirrus.

**[Done]**

Figure 12&13: Why the question mark in the title for figure 12? Is this an oblique reference to the possible interference at 504 nm? Can you speculate on the departure from a nearly straight Angstrom relationship between 375-450 nm? Why not condense figures 12 & 13 into one figure keeping the log-log and axes limits from figure 13? It looks like both 870 nm and 1020 nm fall outside 0.008 AOD limits.

**["?" removed; Figures 12 and 13 now one figure; Speculated that dip was associated with difficulty in determining Vos with the high spectral resolution and low signal to noise at those wavelengths.]**

Technical corrections:
Figure 4: If you're not going to show units on the y-axis, you may as well hide the numbers as well or normalize to unity. But I'd rather see the units. And it would be very interesting to see R [from the calibrated lamp irradiance] and Ro[from Langley Vo tied to ET irradiance] in this same figure.

**[Units added]**

Figure 5: 1. Put on same scale as figures 3 & 4.

**[I didn't understand this comment so no action.]**

2. Eliminate ET, not necessary or useful. Also ET doesn't have the same units as either Vo or R.

**[Added text in caption to explain.]**

3. Would be much better to plot responsivities instead of Vo. For one thing, it will avoid confusion from Fraunhofer lines. The two responsivities will only differ significantly in shape in WL regions with gas absorption.

**[I am trying to show reasonableness of interpolation method for Vo that is needed for all derived quantities.]**

Figure 6,7,8: Replace y axis label of "transmission" with "normalized by ET". Fix caption to avoid referring to global and diffuse components as "transmission" because they're not.

**[Done]**

Figures 9,10,11: As above. Also, since dni=0, dhi is equal to ghi, so you can plot either dhi or ghi and elimnate the other two. Then, condense figures 9, 10, 11 into one figure to better show the similarity on days with water clouds and contrast to the day with cirrus.
Figure 12&13: Why the question mark in the title for figure 12? Is this an oblique reference to the possible interference at 504 nm? Can you speculate on the departure from a nearly straight Angstrom relationship between 375-450 nm? Why not condense figures 12 & 13 into one figure keeping the log-log and axes limits from figure 13? It looks like both 870 nm and 1020 nm fall outside 0.008 AOD limits.

**[These were corrected as shown earlier.]**

---

## Author Comment (AC2)

Comment on amt-2021-162
Anonymous Referee #2

This paper reports on the latest version of the Rotating Shadowband Spectroradiometer (RSS) which operates at the ARM site in Oklahoma making spectrally resolved measurements of downwelling solar irradiance in the 360 to 1070 nm wavelength range. The instrument has been upgraded to eliminate deposition on the detector array that degraded the radiometric performance. Additionally, the work describes a technique to radiometrically calibrate spectral regions of strong absorption (e.g. oxygen and water vapor) where Langley analysis to determine the top-of-atmosphere irradiance (the so-called V0) is not valid. The paper also reports on the correction of the spectral calibration (registration) with the use of solar atmospheric absorption features. Finally example spectra of: direct normal, global and diffuse horizontal, transmission, aerosol optical depth spectra are presented. I have several comments and suggestions that should be addressed prior to the publication of this work.

Major Comments:
1) The section 3.2 closely follows the technique demonstrated by Kindel et al (Applied Optics, 2001) using a hybrid of Langley channels in the spectral regions where transmission follows Beer's law and a standard lamp to fill in the regions of strong molecular absorption. In that work, a wavelength independent scaling factor was used to account for geometry and bring the calibration coefficients from Langley and standard lamp into agreement. This had the dual purpose of filling in spectral regions where Langley is invalid (as is done in this work) and demonstrating, to level of uncertainty of TOA solar spectra, about 2-3% and at the time of publication, the agreement between the two methods (standard lamp and Langley) for radiometric calibration of solar spectroradiometers. I am surprised, given the similarity of the technique described here that no mention or description of that work is given. It should be.

**[The authors apologize for missing this reference: This paragraph was added: "Kindel et al. (2001) describe a technique for determining extraterrestrial (ET) responses over the continuous solar spectrum using spectral regions where Langley analyses are appropriate and lamp calibrations where they are not. The lamp and Langley calibrations where Langley calibrations are valid determine a scale factor applied to the lamp calibrations that is used to estimate ET responses in the spectral regions where Langleys are not appropriate. In this paper we used the method described below because we missed reading this paper."]**

2) Langley analysis is usually performed at high altitude sites where relatively pristine atmospheric conditions prevail. The Mauna Loa Observatory is often used in the community for just such calibration experiments. No doubt, this instrument is fixed at the ARM site and the authors are left with using the ARM site for the Langley calibration. This will introduce higher uncertainties. It is impossible to determine what those uncertainties might be from this paper. Ideally, the V0s should be stable over the course of many Langley plots. A stable instrument, combined with good Langley conditions will reproduce the same V0s from Langley plot to Langley plot. To give the reader an understanding of just how good the instrument and the ARM site are for Langley analysis I recommend plotting the V0s with their standard deviations. I worry that the aerosol environment of the ARM site is less than ideal. As the authors point out, creating a valid (modified) Langley plot for water vapor requires that water vapor is stable over the course of the Langley plot measurement time frame. This is equally true of aerosols, although likely not as problematic as water vapor. Additionally, the V0s generally have higher uncertainties in the shorter wavelengths. The higher optical depths likely encountered at the ARM site will generate larger slopes that create larger changes in the V0s when extrapolated to the TOA.

**[This paragraph added with references to explain why we think Oklahoma data are okay: "While Langley calibrations are ideally performed at high altitude sites, such as Mauna Loa Observatory (MLO), we have demonstrated that calibrations can be performed at less than this ideal site using a sufficient number of Langleys. Perhaps the best example of this is from Michalsky and LeBaron (2013). Using the same multifilter rotating shadowband radiometer (MFRSR) at MLO and in Boulder, Colorado, we obtained the same calibration in the five aerosol channels of the MFRSR to within 0.006 worst case. Granted, the signal to noise was much better at MLO even with fewer Langley events, but the medians at both places agreed (see Figure 1 in Michalsky and LeBaron (2013)). Note that the MFRSR and the RSS use the same shadowing procedure in every respect, therefore, we expect similar results from either instrument."]**

3) The authors carefully identify the source of nearly every absorption feature in the spectra shown. However, there is an effect that is not discussed at all. The so-called planetary problem. This is the result of multiple scattering between the surface and the atmosphere. In measurements of downwelling global spectral irradiance taken from aircraft it's common to see the signature of the vegetation near infrared edge (~690-800 nm) when flying below a cloud deck. Indeed, the first surface measurements made with an ASD-FR spectroradiometer and cosine diffuser showed this effect and it remained a mystery until later explained by Dr. Peter Pilewskie who made similar measurements with the Solar Spectral Flux Radiometer. Thankfully, over the ocean this effect is largely absent thus the strong preference for using flight data over ocean for shortwave cloud studies. I strongly suspect that this type of effect on the spectral shape of the irradiance can be found in RSS data. I understand that it's extremely hard to quantify without good measurements of surface spectral albedo and even then I'm not sure how those measurements would be used to correct the data. The authors should, at the very least, acknowledge the unknown continuum effect that the surface albedo has on the RSS measurements, particularly in the presence of clouds, perhaps showing a plot of a global irradiance spectrum obtained during peak growing season under cloud cover. The feature is generally easy to identify.

**[We thank this referee for reminding us of this effect; this was added: "This increase in transmission, outside molecular bands is likely caused by the increasing surface albedo with wavelength (see, e.g., McFarlane et. al. 2011), which increases the shortwave radiation scattered to the clouds, hence, causing them to appear brighter in the near-infrared."]**

4) The comparison of aerosol optical depths from RSS and the well-established Cimel instrument (Figure 13) are very useful and give confidence to the results obtained by RSS. However, a single comparison, as good as it is, it not useful. Surely, this figure can be expanded so that statistics can be calculated to see how well the two instruments agree. Perhaps aerosol optical depths from both instruments on those days used for the Langley analysis could be included?

**[We added references to earlier papers that have shown this comparison for shadow banding instruments that work in exactly the same mode.]**

5) There is no summary/conclusions section to this work. I often find this, after the abstract, to be the most important section in any scientific paper.

**[Now there is.]**

Minor Comments:
1) The use of Kurucz/Gueymard is outdated. The TSIS Hybrid Solar Reference Spectrum has uncertainties substantially below 1% for most of the wavelength range discussed in this work. (Coddington, 2021). I understand that this comes too late for this work, but I would encourage the authors to reference this work as a potential fix for their comment on line 265 about the TOA solar spectrum uncertainty.

**[Added text indicating new ET spectrum.]**

 Additionally, the Kurucz spectrum is scaled by Gueymard. It's not clear how. Is it normalized at a single wavelength or by their integrated values?

**[Not sure how to improve statement.]**

2) Line 207-208, I'm not sure what is meant by the sentence "The feature labeled H2O(?) is likely a water vapor band that is in the HITRAN (2012) database". The feature is in the spectrum and I assume all the features described in the paper are in the HITRAN database or atlases of solar spectra. Please reword.

**[Uncertain wording removed and now stated that it is water vapor as in HITRAN.]**

3) Line 213-214. "... is the broad depression in what looks like continuum". Chappuis absorption doesn't look like continuum, it is continuum. Reword.

**[Reworded.]**

4) Line 122 (Kindel, 2001) used synthetic Langley plots created with MODTRAN to show exactly where in the spectrum(400-2500 nm) Langley (i.e. Beer's Law) is valid.

**[This was added.]**

5) Line 233, please explain how Rayleigh and Chappuis were removed, it has a large effect on the values of the aerosol optical depth. On line 183, the retrieval of ozone from the Chappuis is "less obvious and more uncertain". Certainly this applies to its removal to determine aerosol optical depth as well?

**[References to how it was done were added.]**

6) The transmission uncertainty is estimated at 1% on line 265. Following the comments above about the lack of uncertainty analysis in the generation of the V0s it's impossible to assess this claim.

**[Reference to how this determined for spectrum not affected by molecular absorption added.]**